# Computational analysis of tongue reconstruction surgery: The impact of donor site and flap volume on post-operative anatomy and biomechanics

Amir Reza Isazadeh[1]*, Lindsey Westover[2], Hadi Seikaly[3,4], Daniel Aalto[1,4]

**1** Department of Communication Sciences and Disorders, Faculty of Rehabilitation Medicine, University of Alberta, Edmonton, AB, Canada, **2** Department of Mechanical Engineering & Department of Biomedical Engineering, Faculty of Engineering, University of Alberta, Edmonton, AB, Canada, **3** Division of Otolaryngology, Department of Surgery, Faculty of Medicine and Dentistry, University of Alberta, Edmonton, AB, Canada, **4** Institute for Reconstructive Sciences in Medicine (iRSM), Misericordia Community Hospital, Edmonton, AB, Canada

* isazadeh@ualberta.ca

## Abstract

Tongue reconstruction requires a series of decisions tailored to patient needs to restore anatomy and preserve speech and swallowing. The impact of these interdependent choices is difficult to evaluate, as clinical outcomes depend on case-specific factors. However, computational analysis offers a method for analyzing these interdependencies in a controlled fashion. The present study systematically quantifies the impact of key surgical decisions, namely donor site selection (radial forearm vs. anterolateral thigh) and flap volume on the final anatomical and biomechanical outcomes. To achieve this, we developed an automated framework that simulates free flap tongue reconstruction. The framework leverages biomechanically optimized flap design to generate a multi-component virtual flap, which is then computationally sutured to the resection site, and its long-term tissue atrophy is simulated to predict the final neotongue state. Using this platform, we conducted a 120-simulation factorial study for the systematic analysis. Four clinically plausible tongue reconstruction scenarios, six levels of flap stiffness, and five levels of flap overbulking (intentional excess volume) were simulated. The results reveal a fundamental, physics-based trade-off: while increasing flap overbulking was the dominant factor in restoring pre-operative anatomy, it came at the cost of a predictable increase in biomechanical strain imposed on the native tongue. Furthermore, stiffer flaps induced significantly higher baseline strain. The anatomical benefit of overbulking was significantly modulated by tissue properties. These findings provide a biomechanical rationale for clinically observed functional trade-offs. This work presents an open-source, physics-based, and robust computational testbed for systematically evaluating interdependent surgical variables. Ultimately, the framework's automation and scalability

**Data availability statement:** The simulation framework is made openly available under MIT license, without restriction, enabling full reproducibility of all 120 simulations and direct application to new patient cases. The "Raw" MRI images cannot be publicly shared due to privacy restrictions imposed by the Health Research Ethics Board of Alberta. However, all inputs required to reproduce the simulation outcomes (including the segmented geometries) and statistical results are included in this repository (hosted on GitHub at https://github.com/is-Amir/VSP-BioSim and archived with a citable DOI: 10.5281/zenodo.17438091). Access to de-identified MRI data may be granted to researchers upon request to the corresponding author or the Health Research Ethics Board of Alberta – Cancer Committee (cancer@hreba.ca).

**Funding:** This work was supported by the Alberta Cancer Foundation (grant number 27 601) awarded to DA. The funder had no role in study design, data collection and analysis, decision to publish, or preparation of the manuscript.

**Competing interests:** The authors have declared that no competing interests exist.

offer a pathway toward personalized, simulation-informed surgical planning for tongue reconstruction.

## Introduction

Surgical removal of the cancerous defect from the tongue (glossectomy) is a primary treatment for tongue cancer. The standard of care is microvascular free-flap reconstruction, which involves transplanting tissue to rebuild the tongue. However, this procedure can severely compromise speech and swallowing [1]. Successful reconstruction relies on interdependent surgical decisions tailored to the individual. The surgeon's strategy is influenced by the defect's characteristics, including size, volume, location, and the geometry of the resection. Key considerations involve donor-site selection and flap design (both planar geometry and volume) to restore anatomy [2,3]. While research has addressed patient-specific planar geometry via template-based [3–5] or algorithmic approaches [6,7], other critical decisions remain largely guided by experience.

Preserving residual tongue mobility is the foremost determinant of functional recovery; mobility for speech and swallowing precedes bulk. Although alternative donor sites, such as the rectus abdominis (RA) and latissimus dorsi (LD), are used for massive defects, the radial forearm free flap (RFFF) and anterolateral thigh (ALT) flap are predominant in practice, representing the biomechanical spectrum from thin, pliable tissue to substantial bulk [1,2,8,9].

The RFFF is often preferred for defects where pliability and conformity are paramount: it is thin, pliable, and readily contours to the mobile anterior tongue, supporting articulation [8,10,11]. Conversely, the ALT flap is favored when substantial bulk is needed, particularly for base-of-tongue defects where palate contact is essential for swallowing, or when forearm donor-site morbidity affecting wrist function should be avoided, as many patients prioritize hand function and cosmesis [8,9,12,13]. This entails a trade-off: the stiffer, bulky ALT flap can tether the tongue, impairing articulation, whereas the compliant RFFF mitigates tethering but may provide insufficient long-term volume for swallowing [10,13]. This decision is a complex, experience-based judgment on long-term consequences. While studies have investigated functional outcomes [8,9,13], assessing the impact of donor-site selection and flap volume on post-operative anatomy and biomechanics provides granular insights.

A universal challenge in reconstructive surgery is the significant post-operative volume loss of free flaps (atrophy), driven by denervation, muscle disuse, and fat resorption. To compensate, surgeons employ "overbulking," harvesting a flap larger than the defect. However, determining the optimal overbulking volume remains largely subjective because evidence-based recommendations are limited to a handful of studies and depend on individual circumstances [14–19]. The consequences of miscalculation are clinically significant: undercorrection results in inadequate volume for palate contact, impairing speech and swallowing, while excessive overbulking produces a bulky, immobile tongue that obstructs the airway and hinders articulation [20].

The interdependent choices of donor-site selection and flap overbulking are highly patient-specific, often necessitating reliance on intra-operative experience. Donor-site selection determines not only the available tissue volume but also the degree of biomechanical mismatch between the flap and the native tongue. Similarly, the extent of overbulking must account for a complex array of variables, including resection geometry, BMI, patient age, adjuvant radiation therapy (RT), and anticipated tissue atrophy [17,18]. These decisions are inherently coupled, creating a complex, case-specific decision matrix that is difficult to navigate without quantitative guidance.

Systematically isolating these variables via prospective randomized clinical trials faces substantial ethical and methodological hurdles [21,22]. Ethically, randomizing patients to surgical arms that are potentially suboptimal for their specific defect is indefensible. Methodologically, the high dimensionality of confounding factors, ranging from tumor morphology and patient physiology to diverse treatment parameters, introduces significant variability that obscures the causal effects of surgical decisions. While theoretically manageable through large-scale, multi-center studies, the requisite sample size and logistical complexity render a traditional clinical investigation an impractical initial approach.

Biomechanical modeling offers a viable and ethical alternative to investigate these variables and gain insights into their interactions. Buchaillard and colleagues [23,24] pioneered this by utilizing a generic finite element model to simulate hemiglossectomy and floor-of-mouth reconstruction, modifying tissue stiffness in affected areas to approximate surgical outcomes. Although simplified, this approach provided functional insights aligned with clinical observations. Subsequent studies refined this foundation by introducing multi-body dynamics, rigid-soft tissue interactions, advanced constitutive models, and various muscle modeling techniques to capture the complex biomechanics of the oropharynx with greater fidelity [25–28]. To streamline patient-specific modeling, Harandi et al. [29,30] developed registration methods to adapt generic models to individual MRI anatomy. Kappert et al. [31] further advanced this by incorporating detailed muscle fiber orientations from diffusion-weighted MRI (DW-MRI). While these techniques are effective for assessing patient-specific functional nuances, further refinements are needed to fully translate them into robust surgical intervention analysis.

To our knowledge, few studies have simulated surgical interventions on the tongue. Foundational work modeled the functional consequences of glossectomy by altering the material properties of affected regions [23]. This method of modifying regional soft-tissue properties was subsequently adapted to investigate the biomechanical impact of radiation-induced fibrosis [32]. In a broader context, analogous computational frameworks have been established to model bioheat transfer and laser-induced thermal damage in living tissues [33–36]. Moving beyond material-property modification alone, Kappert et al. [37] introduced a framework for simulating small resections treated with primary closure. Their approach computationally replicated suturing by pulling resection edges together, updating the finite-element geometry, and locally increasing stiffness to mimic scarring.

Building on these foundations, this study creates an end-to-end, patient-specific simulation framework in which the free flap is modeled as an independent, multi-component body virtually sutured to the resected tongue. This approach faithfully replicates the surgical workflow, including suture attachment and tensioning. Furthermore, it incorporates a biological phenomenon, tissue atrophy, to estimate the final post-remodeling anatomical and biomechanical state.

The proposed robust, automated workflow establishes a computational testbed for systematic surgical analysis. Leveraging this framework, we conducted a factorial experiment to decouple and quantify the impacts of two pivotal surgical decisions: donor-site selection and flap overbulking. By systematically varying flap biomechanics and subcutaneous volume, we isolated the independent and interactive effects of these variables on the outcomes. This modeling approach generates quantitative, physics-based insights into complex surgical trade-offs that are difficult to disentangle through clinical studies alone.

## Materials and methods

### Patient-specific tongue model

This study is built upon a high-resolution, patient-specific 3D model of the tongue, previously segmented using Mimics Medical 17.0 (Materialise, Leuven, Belgium) from an MRI scan of a healthy male participant (BMI: 30.4 kg/m$^2$) in his 40s

[7]. Data acquisition was approved by the Health Research Ethics Board of Alberta (Protocol HREBA.CC-17–0597), with written informed consent obtained from the participant upon recruitment on 30/07/2019. The tongue model establishes the baseline pre-operative anatomy for all simulations herein. To span a representative spectrum of Urken class I–II tongue defects [2], four resections were computationally simulated as detailed previously [7] and described in Fig 1. For each case, the "resected tongue model" was generated via a Boolean subtraction of the corresponding resection volume from the baseline tongue anatomy using Blender (Blender Foundation, Amsterdam, Netherlands). The resulting volumetric mesh serves as the recipient site for the virtual free flap reconstruction, and the resection model provides the geometric basis for subsequent flap design (Fig 1).

To establish a stable anatomical reference frame and define physiological boundary conditions, a rigid jaw model was also created. The mandibular surface was first segmented from the patient's MRI data using Mimics Medical. Subsequently, a high-resolution generic jaw template was registered to this surface using an elastic mesh deformation algorithm

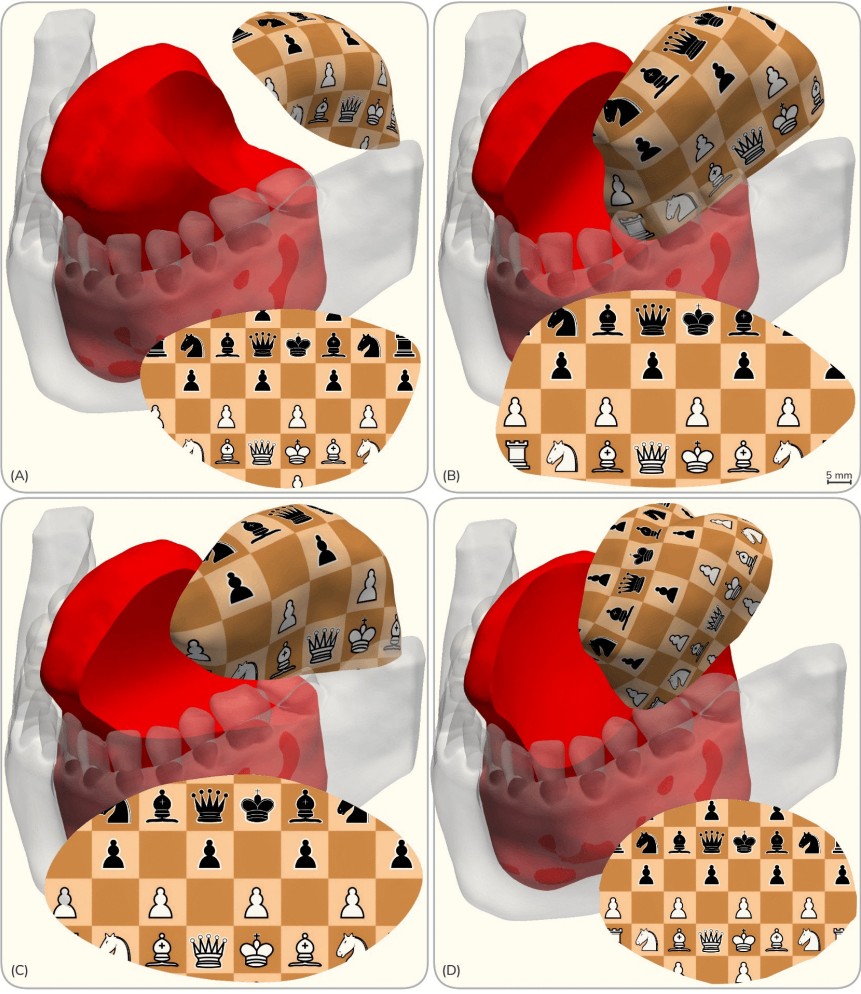

**Fig 1. Patient-specific tongue model, resection scenarios, and flap designs.** Overview of the four simulated tongue cancer cases: (A) tip-sparing hemiglossectomy; (B) lateral hemiglossectomy; (C) posteriorly extended hemiglossectomy sparing the tongue base; and (D) asymmetric, midline-crossing partial glossectomy. The biomechanically optimized planar flap designs are generated using the hiFEM algorithm, as detailed in previous work [7]. The texture maps each point in the resection model to its corresponding point in the planar flap design.

in Blender, guided by manually placed anatomical landmarks. The resulting registered jaw model serves as a rigid, stationary foundation for the simulation and defines the anatomical locations of the tongue's muscular attachments.

## Biomechanically optimized flap design

As an input for the surgical simulation, a patient-specific three-dimensional free flap model is generated from the resection model (Fig 2A), which is composed of skin and adipose layers (Fig 2B–2D). The process begins by employing the hiFEM (hyperelastic inverse Finite Element Method), as detailed in previous work [7,38], to produce a biomechanically optimized planar skin template for each resection case (Fig 2C). A key challenge, however, is the unknown geometric model of the subcutaneous adipose tissue.

Mechanically, adipose tissue exhibits a very low shear modulus, allowing it to undergo large, quasi-fluidic deformations with relatively low elastic energy storage [39]. Its primary contribution to the flap's integrity is volumetric bulk rather than a predefined geometric shape. This principle is clinically leveraged in techniques like "beavertail" augmentation, where supplementary adipose tissue is malleably shaped to increase flap volume [10]. Consequently, the critical modeling

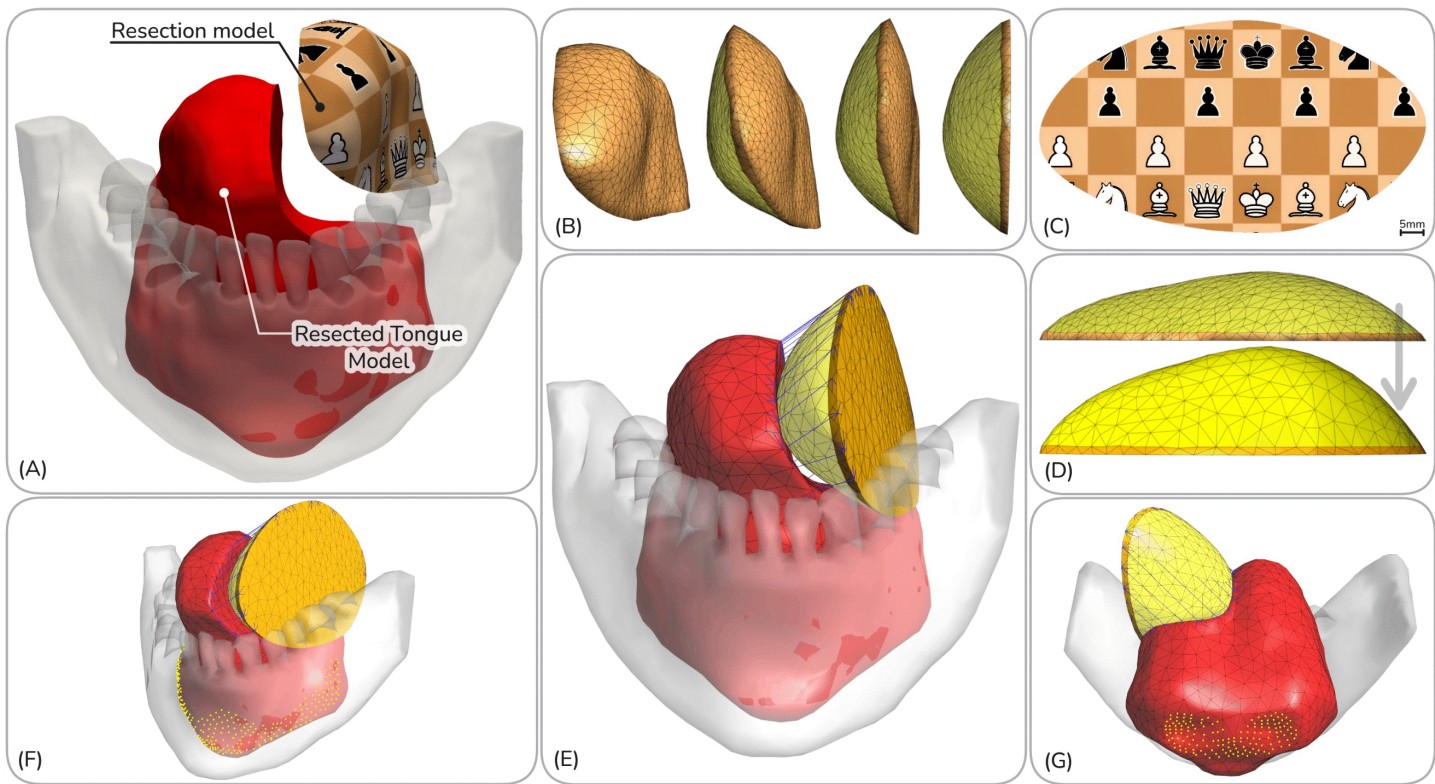

**Fig 2. Workflow for free flap model generation and surgical simulation pre-processing.** Panel A shows the two primary inputs: the "Resected Tongue Model" (red) and "Resection Model" (textured). First, the hiFEM algorithm creates an optimized 2D planar skin template for the flap (C: flap's frontal view). This template's deterministic 3D-to-2D mapping then guides the computational "unfolding" of the resection model to generate a multi-component free flap model (B: flap's side view). To simulate the clinical practice of "overbulking," the flap's volume is increased by scaling the thickness of its yellow adipose component (D: flap's top view). Next, the multi-component flap is positioned at the resection site (E), and one-dimensional tension elements (blue lines) are defined to connect corresponding suture points on the flap and the resected tongue. Anatomical boundary conditions are established by fixing nodes (yellow dots) that represent the tongue's muscular attachments to the mandible and hyoid bone, shown in anterior (F) and posterior (G) views.

requirement is to define an initial adipose geometry that has the necessary volume and maintains geometric congruency with the planar skin template.

To achieve this, a reproducible initial 3D shape for the adipose layer is generated by computationally "unfolding" the original 3D resection volume (Fig 2B). This transformation is guided by the deterministic 3D-to-2D nodal mapping generated by the hiFEM solution. This step ensures the resulting adipose layer is perfectly congruent with the planar design from the hiFEM algorithm and creates a geometrically consistent foundation for assembling the composite flap model. The multilayer flap model is discretized with a high-quality volumetric tetrahedral mesh using fTetWild [40], which guarantees congruent meshing and a seamless interface between the skin and fat domains.

## Simulation of tongue reconstruction surgery

The surgical procedure of tongue reconstruction was computationally simulated using the open-source finite element solver FEBio [41] to establish the post-operative neotongue configuration (Fig 3i–3iii). This involves insetting the multi-component flap onto the resected tongue and applying sutures to close the defect. First, a series of suture points is established along the flap's perimeter, spaced at clinically relevant 5 mm intervals [42]. To establish a corresponding set of attachment points on the resected tongue model, the simulation leverages the deterministic 3D-to-2D nodal mapping generated by the hiFEM solution. This critical step ensures that the boundary deformation is guided by the optimized geometry derived from the flap design stage.

A multi-step numerical approach was developed to simulate the application of sutures and ensure a realistic distribution of forces. To mimic how a physical suture engages a small area of tissue and avoid artificial stress concentrations, the force at each suture point was applied to a small cluster of nodes within a 2 mm radius rather than to a single node. The force is distributed to the cluster nodes via linear-spring elements that connect all nodes in a cluster to a central point. The corresponding central points on the flap and the resected tongue are then drawn together using time-activated, one-dimensional tension elements that shorten to a near-zero length over the simulation step. This sequence computationally replicates the surgical action of tightening and securing sutures.

Following the virtual suturing, a subsequent step allows the system to equilibrate for five minutes. This duration was quantitatively chosen by evaluating the decay of the shear relaxation modulus, $G(t)$, using the viscoelastic parameters

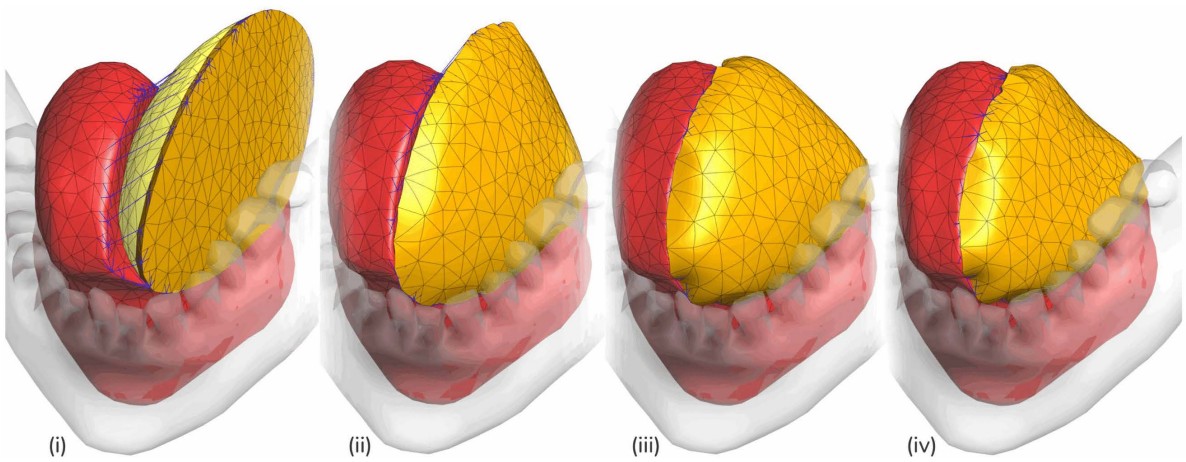

**Fig 3. Surgical simulation pipeline: reconstruction to long-term remodeling.** (i) Initial simulation setup, (ii) The suture model brings the flap proximal to the resected tongue model and simulates suturing. (iii) After the 5-minute idling for stresses to relax with the viscoelastic model present, this is "Day 0" immediately after reconstruction. (iv) shows the tongue model after one year of post-surgery atrophy modeling.

detailed in Table 1. The analysis confirmed that >95% of the total transient stress relaxation had completed within this interval (97.1% for tongue, 97.8% for skin, and 100% for adipose tissue). By modeling this phenomenon, internal stresses are allowed to redistribute throughout the neotongue. The final, equilibrated geometry after this relaxation represents the immediate post-operative "Day 0" anatomical state (Fig 3iii), which provides a stable configuration for the subsequent simulation of biological remodeling.

## Simulation of post-operative flap atrophy

A critical challenge in free flap tongue reconstruction is the significant, time-dependent volume loss that occurs post-operatively. Clinical studies have reported a wide range of atrophy, with long-term volume reduction spanning from approximately 20% to nearly 60% [14,17,18,52]. The final flap volume is heavily influenced by factors such as tissue composition, patient age, BMI, and most critically, the administration of adjuvant radiation therapy. For instance, at one year post-operatively, ALT flaps in patients receiving radiotherapy exhibited an average volume loss of 44.2%, compared to just 19.8% in non-irradiated patients [14]. Similarly, radial forearm free flaps (RFFF) have shown a mean volume loss of 42.7% at ten months post-radiotherapy [17]. Given that radiotherapy is a common component of treatment for advanced cancers, and considering this body of evidence, a total volume loss of 40% was selected for this study as a clinically representative value to model the extent of flap atrophy over a one-year period. While this value serves as a clinically representative average for this systematic study, its application to an individual patient would require personalization based on known risk factors for atrophy, such as the planned radiotherapy dose, patient age, and BMI.

To simulate this phenomenon, the model is evolved from its "Day 0" state (Fig 3iii) to an estimated one-year post-operative state (Fig 3iv). This remodeling was implemented by applying a time-dependent, isotropic negative kinematic growth to the finite elements of the flap's subcutaneous tissue. Consistent with clinical observations, atrophy was confined to the flap; the native tongue was assumed to maintain its volume. The simulation progresses over the defined one-year timeframe, with appropriate time-scaling to ensure computational tractability. The final, equilibrated geometry represents

**Table 1. Material parameters used in the finite element simulations.**

| Soft Tissue | Model | Parameters (Values) | Source |
|---|---|---|---|
| Native Tongue | Yeoh (Hyperelastic, $N=2$) | $(c_1, c_2, k) = (1.037kPa, 0.486kPa, 100kPa)$ | [24] |
| Native Tongue | Prony (Viscoelastic, $N=4$) | $(\tau_1, \tau_2, \tau_3, \tau_4) = (0.05s, 1.0s, 20s, 400s)$; $(\gamma_1, \gamma_2, \gamma_3, \gamma_4) = (39.0, 4.526, 4.895, 3.211)$ | [43][†] |
| Subcutaneous Adipose | Ogden (Hyperelastic, $N=1$) | $(c_1, m_1, k) = (9.0kPa, 6.0, 100kPa)$ | [44] |
| Subcutaneous Adipose | Prony (Viscoelastic, $N=3$) | $(\tau_1, \tau_2, \tau_3) = (0.1s, 1.0s, 10.0s)$; $(\gamma_1, \gamma_2, \gamma_3) = (4.428, 8.692, 0.738)$ | [45][†] |
| forearm skin (stiffness levels) | Ogden (Hyperelastic, $N=2$) | Level 1: $(c_1, c_2, m_1, m_2, k) = (41.5kPa, 1.39Pa, 1.223, 41.672, 200kPa)$; Level 2: $(c_1, c_2, m_1, m_2, k) = (61.3kPa, 4.38Pa, 7.059, 39.062, 200kPa)$; Level 3: $(c_1, c_2, m_1, m_2, k) = (102.1kPa, 5.20Pa, 7.660, 38.691, 200kPa)$ | [‡] |
| thigh skin (stiffness levels) | Ogden (Hyperelastic, $N=2$) | Level 1: $(c_1, c_2, m_1, m_2, k) = (90.0kPa, -10.0Pa, 13.431, 31.000, 200kPa)$; Level 2: $(c_1, c_2, m_1, m_2, k) = (180.0kPa, -60.0Pa, 14.531, 36.333, 200kPa)$; Level 3: $(c_1, c_2, m_1, m_2, k) = (360.0kPa, -120.0Pa, 13.839, 35.274, 200kPa)$ | [‡] |
| Skin | Prony (Viscoelastic, $N=4$) | $(\tau_1, \tau_2, \tau_3, \tau_4) = (0.3791s, 4.1987s, 6.9961s, 180.02s)$; $(\gamma_1, \gamma_2, \gamma_3, \gamma_4) = (3.649, 0.452, 0.553, 0.627)$ | [46][†] |

[†]The referenced studies define the relaxation function differently from Eq 3. The reported viscoelastic parameters have been mathematically converted to be consistent with the formulation used in this study. Relaxation coefficients ($\gamma_i$) are dimensionless.

[‡]Material profiles for the three stiffness levels of forearm skin were calibrated using experimental data from several published studies [47–50]. Similarly, the profiles for thigh skin were calibrated based on data reported by [51].

the estimated one-year anatomical and biomechanical state of the neotongue, and this 3D model serves as the primary output for the subsequent quantitative analysis.

## Boundary conditions and contact mechanics

A stable and physiologically realistic reference frame was established by defining the anatomical boundary conditions. The jaw model was treated as a rigid body and was fully constrained in all degrees of freedom. The tongue's physiological connection to this frame was modeled by fixing the nodes of the tongue mesh at its anatomical attachment points to the mandible and hyoid bone, which include the genioglossus, mylohyoid, and hyoglossus muscle origins as illustrated in Fig 2F and 2G. This configuration ensures that all simulated deformations originate from a stable anatomical base. As the objective was to estimate the final, stable anatomy at rest, muscle activation was not included in the model; however, gravity was applied as a global body force.

The physical interface between the flap and native tongue was modeled using a robust contact formulation. A penalty-based contact algorithm was enforced between the surfaces of the flap and the resected tongue to prevent interpenetration. To computationally model the desired surgical outcome of complete tissue apposition, a "sticky" contact formulation was also specified at the interface of the flap and the resected tongue surface. The simulation results confirmed that this approach prevented the formation of gaps or voids, and yielded a seamless integration at the tongue-flap interface. Additionally, a penalty contact was defined between the tongue and the jaw model to prevent interpenetration.

## Constitutive models

All soft tissues (tongue, skin, and subcutaneous adipose tissue) were modeled as nearly incompressible, isotropic, non-linear, hyperelastic, and time-dependent materials. The native tongue was modeled using the Yeoh hyperelastic model, while the donor flap skin and adipose tissue were modeled using the Ogden hyperelastic model, following the respective FEBio definitions as shown below [41].

$$\Psi_{\text{Yeoh}} = \sum_{i=1}^{N} c_i \left( \tilde{I}_1 - 3 \right)^i + \frac{k}{2} (\ln J)^2$$

(1)

$$\Psi_{\text{Ogden}} = \sum_{i=1}^{N} \frac{c_i}{m_i^2} \left( \tilde{\lambda}_1^{m_i} + \tilde{\lambda}_2^{m_i} + \tilde{\lambda}_3^{m_i} - 3 \right) + \frac{k}{2} (\ln J)^2$$

(2)

where $\Psi$ represents the strain energy density function. In both models, the volumetric response is governed by the material's bulk modulus, $k$, and the volume ratio, $J$, which is the determinant of the deformation gradient tensor. $N$ denotes the number of terms in the summation. The specific values of $N$ used for each tissue type are detailed in Table 1. The Yeoh model's isochoric behavior is a function of the first deviatoric strain invariant, $\tilde{I}_1$, where $c_i$ are the material parameters. In the Ogden formulation, the isochoric response is described in terms of the deviatoric principal stretches, $\tilde{\lambda}_i$, where $c_i$ and $m_i$ are material parameters.

To simulate the immediate post-operative behavior, the hyperelastic models were augmented with uncoupled viscoelastic properties to capture stress relaxation. The deviatoric part of the second Piola-Kirchhoff stress, $\mathbf{S}$, is calculated using the following integral:

$$\mathbf{S}(t) = \int_{-\infty}^{t} G(t-s) \frac{d\mathbf{S}^e}{ds} ds, \qquad G(t) = 1 + \sum_{i=1}^{N} \gamma_i \exp\left( -\frac{t}{\tau_i} \right)$$

(3)

where the current stress is determined by the entire history of the elastic deviatoric stress, $\mathbf{S}^e$, derived from the hyperelastic potential. The material's time-dependent behavior is characterized by the relaxation function, $G(t)$, which is represented by a Prony series. In this series, $\gamma_i$ and $\tau_i$ are the relaxation parameters and relaxation times for each of the $N$ terms.

## Soft tissue parameterization

To investigate the impact of donor-site selection, material parameters for forearm and thigh skin were derived from a thorough literature review (Fig 4). Given the substantial stiffness variability observed within and between donor sites, we defined three distinct stiffness levels (low, mid, and high) for each site to encompass the physiological range. For thigh skin, parameters were derived from the statistical data reported by Zhou et al. [51]. The mid-stiffness model was calibrated to the study's mean stress–stretch curve, while the high- and low-stiffness models were calibrated to the mean ± standard deviation (SD) to rigorously bound biological variance. For forearm skin, where experimental observations vary significantly between studies, we captured the average and extremes of reported stiffness. Low-stiffness parameters were calibrated to the most compliant experimental data [47] and high-stiffness parameters to the stiffest reported responses [48,49], with the mid-stiffness level representing the central tendency of the aggregated literature.

To incorporate the observed range of tissue properties into the simulation, a two-term Ogden material model was calibrated to each of the three stiffness levels for both forearm and thigh skin. The calibration was performed using a genetic algorithm optimization procedure implemented in Python using the SciPy library. This yielded six distinct sets of material parameters that encapsulate the clinically relevant range of soft tissue properties (three for each donor site). This methodology enables the factorial study to simulate reconstructions with three levels of tissue stiffness and analyze how tissue biomechanics influence surgical outcomes.

Table 1 summarizes the calibrated hyperelastic parameters for forearm and thigh skin across the three stiffness levels, as well as the constitutive parameters for the other soft tissues used in the simulation. All tissues were assigned viscoelastic properties using a Prony series (Eq 3) to model stress relaxation.

## Simulation stability and robustness

To ensure the robustness and automation required for the 120-simulation factorial study, a strategy was implemented using the open-source finite element solver FEBio, which is specifically designed for biomechanical applications [41]. Significant nonlinearities inherent to the simulation, primarily arising from nonlinear materials and contacts, were managed using an implicit dynamic solver with a robust automatic time-stepping algorithm. The highly nonlinear system of equations was solved using a quasi-Newton (BFGS) method with a nonsymmetric stiffness matrix.

The stability of the simulations was further enhanced through careful model setup. Contact mechanics, a common source of numerical instability, were configured by tuning the penalty parameters to ensure minimal surface penetration without inducing ill-conditioning of the stiffness matrix. To improve the numerical performance for nearly incompressible materials, uncoupled formulations were used for the hyperelastic and viscoelastic constitutive models. A critical component of the strategy was the meshing approach. Given the complex, organic geometries of the tongue and flap, tetrahedral elements were chosen over hexahedral elements due to the availability of automated and robust meshing algorithms, such as the state-of-the-art mesher fTetWild [40]. To mitigate the risk of volumetric and bending locking with linear elements, higher-order quadratic tetrahedral elements (tet10) were used. These elements have been shown to reliably capture complex deformations and contact mechanics without locking artifacts, while providing solution accuracy comparable or superior to hexahedral elements with the same number of degrees of freedom [53].

Finally, a mesh convergence analysis was conducted to ensure that the simulation outcomes were independent of the mesh discretization. Key simulations were repeated with progressively refined mesh densities, and the results were

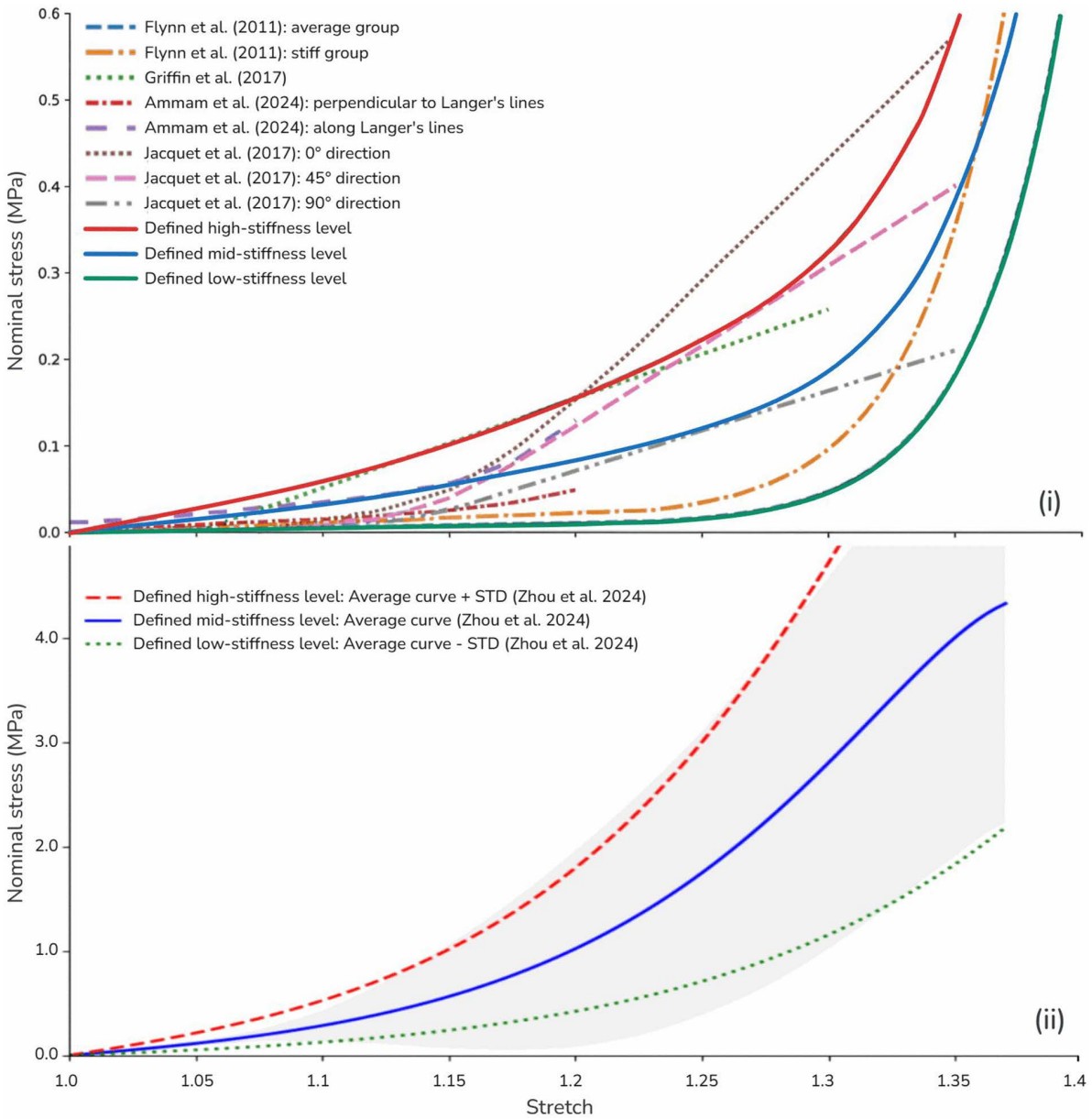

**Fig 4. Literature review of reported donor skin material properties.** (i) forearm skin: The solid lines illustrate the three calibrated stiffness levels—low (green), medium (blue), and high (red)—spanning the range of experimental data reported in the literature (dashed lines [47–50]). (ii) thigh skin: Stiffness levels were derived from Zhou et al. [51]. The experimental mean (mid-stiffness, blue), mean + SD (high-stiffness, red), and mean – SD (low-stiffness, green) were used to capture the physiological range of tissue variability (shaded region).

evaluated using primary outcome metrics: maximum displacement and maximum von Mises equivalent strain. The analysis demonstrated that the solution converged as the mesh was refined. The mesh density selected for the full factorial study was chosen from the converged region to maintain computational tractability. The detailed results of this convergence analysis are presented in S1 Appendix.

## Outcome metrics

To quantitatively evaluate the results of the simulations, two primary metrics were established to assess the final state of the neotongue. These metrics focus on the anatomical fidelity and the biomechanical state of the reconstructed tongue.

The first metric, anatomical state, quantifies how closely the final, post-remodeling neotongue morphology restores the patient's original, pre-operative anatomy. The pre-operative native tongue 3D model serves as the ground truth for this comparison. Since both the simulated and original tongue models are anatomically fixed to the same rigid jaw model, they share a common coordinate system, allowing for direct spatial comparison without the need for further registration. The discrepancy between the two surfaces is quantified by the maximum surface deviation, calculated by the Hausdorff distance. To ensure this metric is robust against numerical artifacts or isolated mesh irregularities, the 99th-percentile Hausdorff distance is reported, providing a stable measure of the largest regional anatomical deviation.

The second metric, imposed tongue deformation, evaluates the biomechanical strain imparted to the native tongue tissue as a result of the reconstruction and subsequent remodeling. To provide a robust and physically meaningful measure of deformation, the von Mises equivalent strain ($\varepsilon_{eq}$) was selected, hereafter referred to as effective strain. This scalar invariant distills the complex, three-dimensional strain state into a single value, permitting a direct and objective comparison across all simulated scenarios. The equivalent strain was calculated from the Lagrange strain tensor, $\mathbf{E}$, as follows:

$$\varepsilon_{eq} = \sqrt{\frac{2}{3}\ \mathbf{dev\,E} : \mathbf{dev\,E}}$$

(4)

where $\mathbf{dev\,E} = \mathbf{E} - \frac{1}{3}\mathrm{tr}(\mathbf{E})\mathbf{I}$ is the deviatoric part of the strain tensor,: denotes the double contraction, $\mathrm{tr}(\mathbf{E})$ is the trace of $\mathbf{E}$, and $\mathbf{I}$ is the identity tensor. This formulation is particularly advantageous as it is based on the deviatoric part of the strain tensor, which mathematically isolates the distortional component of deformation (i.e., shear and tensile strain) from the hydrostatic (volumetric) component. This allows the metric to specifically quantify the mechanically significant distortion that can lead to tissue tethering and functional impairment, while remaining insensitive to the uniform volumetric shrinkage associated with the simulated flap atrophy.

## The factorial study design

A computational factorial experiment was designed to systematically investigate the anatomical and biomechanical impact of key surgical decisions. This approach allows for a controlled analysis of how different factors independently and interactively influence the outcome metrics. The study examined three primary independent variables:

- Patient-Specific Defect Geometry: Four distinct, clinically plausible glossectomy scenarios were used (Fig 1, Cases A–D). These cases represent a range of distinct anatomical defects.

- Donor Site Biomechanics: The mechanical properties of the flap skin layer were varied to simulate tissue harvested from two common donor sites: the radial forearm and the thigh. To account for natural patient-to-patient variability in tissue stiffness, three distinct stiffness profiles were defined for each site.

- Flap Overbulking: Five percentage levels of adipose bulk increase were simulated: 0%, 22.2%, 44.4%, 66.7%, and 88.9%. Overbulking was achieved by scaling the flap's subcutaneous adipose component in the thickness direction (Fig 2D). This approach reflects the clinical practice of elevating a thicker subcutaneous fat layer (for thigh flaps) or harvesting extra beavertail fat tissue (for forearm flaps). This method isolates the effect of flap bulk from its skin paddle size. The selected linear levels span a clinically relevant range, which includes the calculated volume needed to perfectly match the resection size after 40% atrophy (the 66.7% level).

A full factorial design was implemented by combining every level of each factor. This comprehensive approach ensures that all possible interactions between the variables could be studied. The total number of unique simulations conducted was the product of the levels for each factor: (4 defect geometries) × (6 skin stiffness levels) × (5 overbulking levels), resulting in 120 distinct simulation setups.

## Statistical analysis

To quantify the influence of the experimental factors on the simulated outcomes, Ordinary Least Squares (OLS) linear regression was employed for each of the two primary metrics. The independent variables from the factorial design were encoded for the regression models. *Case* was treated as a four-level categorical variable representing the defect geometries, with Case A serving as the reference. *Overbulking* was included as a continuous numerical variable. To decouple the effect of donor site biomechanics, two predictors were created: *is_thigh*, a binary variable (0 = forearm, 1 = thigh), and *stiffness_level*, an ordinal variable coded as {−1, 0, +1} for the three stiffness levels for each donor site. This encoding treats stiffness as a numerical factor and assumes a linear effect for changes in stiffness within a given donor site.

A hierarchical approach was used to build and select the most appropriate model. An initial model included only the main effects of each predictor (Outcome ≈ *Case* + *Overbulking* + *is_thigh* + *stiffness_level*). A second, more complex model was then fitted, containing all main effects plus all two-way, scientifically relevant interaction terms. The final model for interpretation was selected by comparing these nested models using an analysis of variance (ANOVA) F-test. This test determined if the interaction terms provided a statistically significant improvement in explaining outcome variance, with the adjusted R-squared also being evaluated to ensure that added complexity was justified by increased explanatory power.

Following selection, the final model for each outcome was subjected to diagnostic checks to validate the underlying assumptions of OLS regression. The normality of the model's residuals was assessed through visual inspection of quantile-quantile (Q-Q) plots. Multicollinearity was assessed using the Variance Inflation Factor (VIF), ensuring that all VIF scores were below the common threshold of 10 to confirm the stability of the model's coefficients. All statistical analyses were conducted using the *statsmodels* open-source library [54].

## Results

The computational framework demonstrated high stability and robustness, successfully completing all 120 unique simulations in the full factorial study without numerical failures. The simulations provided quantitative outcomes for the final anatomy and biomechanics of the neotongue across the full spectrum of surgical variables tested (Fig 5).

The overall trends are visualized in Fig 6. A clear trade-off emerges between the two primary outcomes: increasing the level of subcutaneous tissue overbulking consistently increased the maximum effective strain imposed on the native tongue (top row), while simultaneously decreasing the maximum surface deviation from the original anatomy (middle row). The final, post-remodeling tongue volume was a direct and linear function of the initial overbulking percentage (bottom row). The 66.7% overbulking level consistently resulted in a final flap volume that closely compensated for the simulated 40% atrophy. At this level, the final tongue volume closely matched the pre-operative native tongue volume, resulting in a percentage change close to zero.

To quantify the factors influencing the first dependent variable (maximum surface deviation), a hierarchical linear regression analysis was performed. A formal model comparison using an ANOVA F-test revealed that the model including two-way interactions provided a significantly better fit to the data than a main effects model ($F = 26.12$, $p < 0.001$). The interaction model explained a substantial portion of the variance (Adj. $R^2 = 0.993$) and was therefore selected for interpretation.

The regression results, detailed in Table 2, identify *overbulking* as the most dominant factor for restoring the original tongue anatomy. The significant negative coefficient (−3.420, $p < 0.001$) indicates that increasing the flap volume consistently reduces the maximum surface deviation. However, the beneficial effect of *overbulking* is significantly modulated by

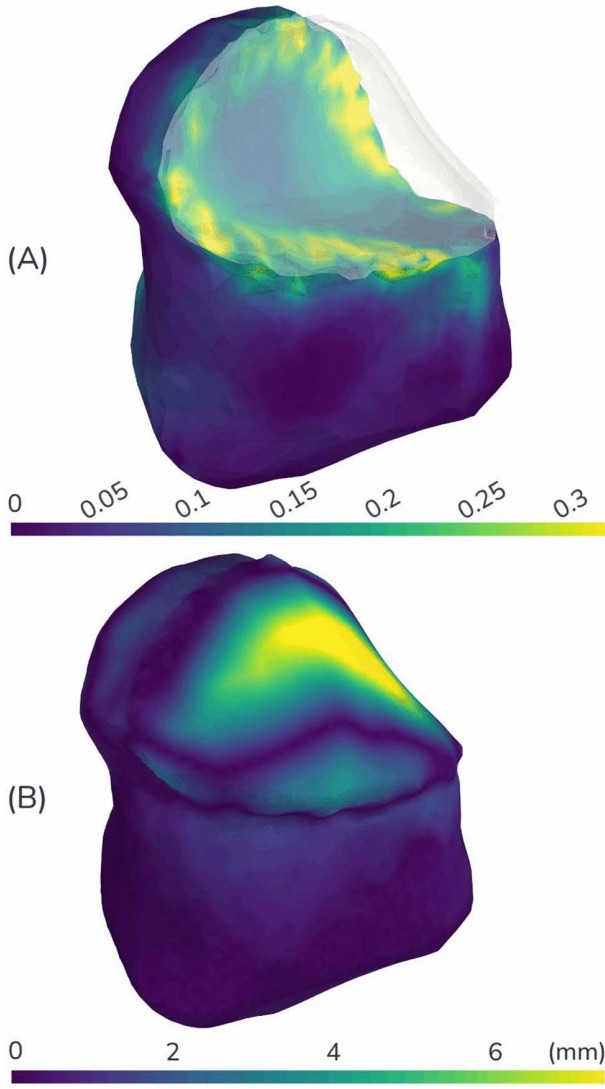

**Fig 5. Representative simulation results showing the spatial distribution of the two primary outcome metrics.** (A) effective strain and (B) surface deviation (Hausdorff distance). These contours capture the end state of the neotongue for a representative reconstruction scenario: Defect Case C reconstructed with a mid-stiffness forearm flap and 66.7% overbulking.

donor site selection (coefficient = 0.494, $p < 0.001$), stiffness level (coefficient = 0.160, $p < 0.001$), and defect geometry (all $p < 0.001$).

A separate regression analysis was conducted for the second dependent variable: maximum effective strain. The model comparison showed that adding interaction terms did not provide any significant improvement over the main effects model ($F = 0.069$, $p = 0.999$). The main effects model was therefore selected as the superior representation of the dependent variable, explaining 86.3% of the variance in maximum effective strain.

The results for the main effects model, presented in Table 3, describe a simple, additive system. All main effects—*overbulking*, *is_thigh*, and *stiffness_level*—were highly significant positive predictors of strain. Increasing flap volume, using tissue from the thigh, and increasing skin stiffness each independently and consistently increased the maximum effective

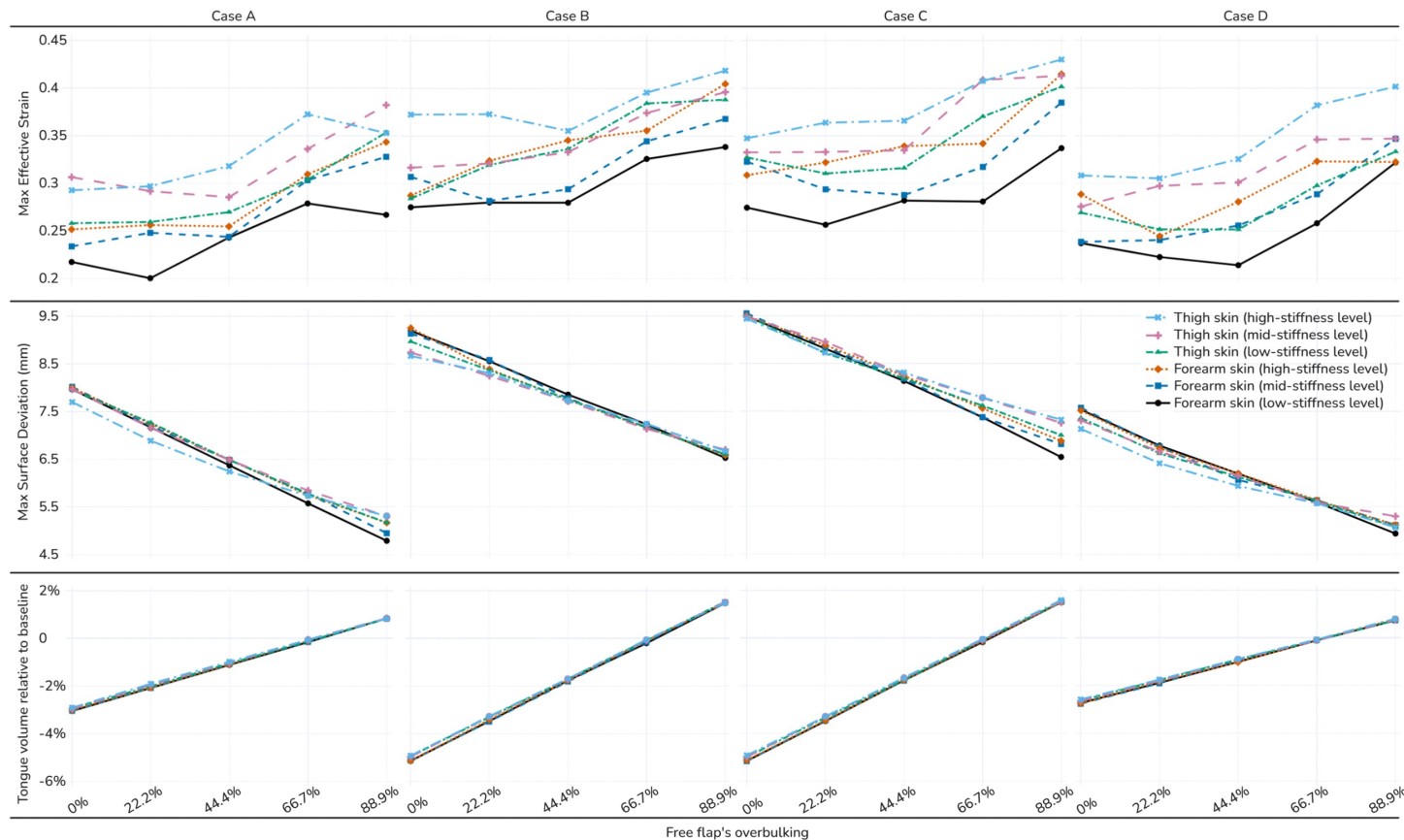

**Fig 6. Visualization of the sensitivity of dependent variables to different skin material properties and different levels of overbulking.** Overbulking increases effective strain (top) while decreasing anatomical deviation (middle); final tongue volume varies linearly with initial overbulking (bottom).

**Table 2. OLS regression results for maximum surface deviation (interaction model). Dependent variable: maximum surface deviation (mm). Reference:** *Case* A; *is_thigh* = 0 (forearm).

| Predictor | Coefficient | Std. Error | t-statistic | p-value |
|---|---|---|---|---|
| Intercept | 8.000 | 0.036 | 220.77 | < 0.001 |
| *Case* B vs. A | 1.101 | 0.046 | 24.01 | < 0.001 |
| *Case* C vs. A | 1.594 | 0.046 | 34.77 | < 0.001 |
| *Case* D vs. A | −0.578 | 0.046 | −12.61 | < 0.001 |
| *overbulking* | −3.420 | 0.067 | −51.38 | < 0.001 |
| *is_thigh* | −0.232 | 0.032 | −7.15 | < 0.001 |
| *stiffness_level* | −0.035 | 0.023 | −1.54 | 0.125 |
| *overbulking* × *Case* B | 0.482 | 0.084 | 5.72 | < 0.001 |
| *overbulking* × *Case* C | 0.335 | 0.084 | 3.98 | < 0.001 |
| *overbulking* × *Case* D | 0.625 | 0.084 | 7.42 | < 0.001 |
| *overbulking* × *is_thigh* | 0.494 | 0.060 | 8.30 | < 0.001 |
| *overbulking* × *stiffness_level* | 0.160 | 0.036 | 4.39 | < 0.001 |
| *is_thigh* × *stiffness_level* | −0.065 | 0.023 | −2.82 | 0.006 |

**Table 3. OLS regression results for maximum effective strain (main effects model). Dependent variable: maximum effective strain. Reference: *Case* A; *is_thigh* = 0 (forearm).**

| Predictor | Coefficient | Std. Error | t-statistic | p-value |
|---|---|---|---|---|
| Intercept | 0.226 | 0.005 | 49.27 | < 0.001 |
| *Case* B vs. A | 0.051 | 0.005 | 10.37 | < 0.001 |
| *Case* C vs. A | 0.052 | 0.005 | 10.67 | < 0.001 |
| *Case* D vs. A | 0.004 | 0.005 | 0.81 | 0.422 |
| *overbulking* | 0.093 | 0.006 | 16.75 | < 0.001 |
| *is_thigh* | 0.043 | 0.003 | 12.32 | < 0.001 |
| *stiffness_level* | 0.023 | 0.002 | 10.74 | < 0.001 |

strain experienced by the native tongue tissue. The defect geometry (*Case*) also had a significant effect, with Cases B and C (larger resections) resulting in higher strain than the reference Case A, while Case D was not significantly different. The lack of significant interactions implies that the strain-inducing effect of *overbulking* was consistent and not modulated by other independent variables.

## Discussion

The fundamental goal of tongue reconstruction is twofold: to restore the anatomical volume required for effective tongue-palate contact during swallowing and speech articulation [10,12], and to preserve the dynamic mobility of the residual tongue essential for chewing, food propulsion, and clear speech [2,8,10]. This study provides a mechanistic and quantitative explanation for the clinical challenges inherent in balancing these objectives. Our simulations reveal a fundamental, physics-based trade-off: surgical decisions aimed at restoring the neotongue's bulk through flap overbulking invariably impose significant and predictable strain on the remaining native tongue tissue. This finding suggests that the "ideal" reconstruction is one that optimally balances volume restoration against the biomechanical cost.

These insights were enabled by the primary methodological achievement of this work: the development of a novel, end-to-end computational pipeline for surgical simulation. This framework uniquely integrates patient-specific model generation, biomechanically optimized flap design, the simulation of tongue reconstruction, and the modeling of biological atrophy. Its power as a robust and automated virtual testbed was exemplified through the successful execution of a 120-simulation factorial study, which enabled such a systematic investigation.

The selection of a donor site is a foundational surgical decision that imposes a distinct and quantifiable biomechanical cost on the reconstructed tongue. Our regression analysis (Table 3) reveals a clear, quantitative relationship: the stiffer the flap tissue, the greater the biomechanical strain imposed on the residual native tongue. Specifically, the model demonstrated that thigh flaps induced a significantly greater mechanical strain on the surrounding native tongue than their forearm counterparts. This biomechanical penalty was further amplified by increasing stiffness levels within each donor site. This finding provides the first physics-based, quantitative measure of the clinical concept of "tethering": the mechanical anchoring of the mobile tongue remnant by a stiff, non-compliant flap.

This quantified tethering effect provides a direct mechanical rationale for the functional trade-offs reported extensively in the clinical literature. Studies consistently associate the thin, pliable radial forearm free flap (RFFF) with superior speech articulation, a function dependent on preserving tongue mobility [8,10,13]. Our results support this clinical observation, demonstrating that the lower material stiffness of the forearm flap results in the lowest imposition of strain, thereby creating a more favorable biomechanical environment for the complex movements of speech. In contrast, for extensive resections creating large volumetric deficits, such as total and subtotal glossectomies and those involving the tongue base, the ALT flap is the preferred reconstructive choice to restore the required bulk [8,9]. Our simulations support the

rationale for this choice but also reveal its inherent biomechanical cost: it simultaneously imposes a greater mechanical strain, which can compromise the mobility required for vital functions.

The clinical practice of overbulking is a direct response to the challenge of post-operative atrophy, yet it forces surgeons into a delicate balancing act to avoid creating a neotongue that is too bulky and immobile [20]. The computational framework provides a clear explanation for this surgical dilemma by quantifying its consequences. The results reveal a fundamental conflict: increasing flap volume is, by a significant margin, the most powerful surgical parameter for reducing maximum surface deviation, which is a direct proxy for restoring the pre-operative anatomy (Table 2, coefficient = −3.420, $p < 0.001$). However, this anatomical gain comes at a direct and unavoidable biomechanical cost, similar to the findings regarding stiffness levels. As shown in Table 3, every increase in overbulking imposes a linear and predictable increase in the strain imparted to the residual native tongue (coefficient = 0.093, $p < 0.001$). This finding quantitatively defines the trade-off between form and function. The "ideal" reconstruction is therefore not simply a matter of perfectly matching the original volume, but rather a patient-specific compromise that optimizes anatomical fidelity against the predictable biomechanical strain that may tether the tongue and impair function—what we know as "too bulky and immobile".

While the biomechanical cost of overbulking is straightforward and additive, its anatomical benefit is highly conditional, modulated by both the properties of the flap and the geometry of the defect. The regression analysis for anatomical deviation (Table 2) is characterized by significant interaction terms, showing that the effectiveness of adding volume is not constant. The most critical interaction is with the donor site (*overbulking* × *is_thigh*, coefficient = 0.494, $p < 0.001$). The positive coefficient reveals that the anatomical benefit of adding volume is significantly less pronounced for the stiffer thigh flaps compared to the more pliable forearm flaps. This finding introduces a key nuance: while a thigh flap shows a minor anatomical advantage at baseline (*is_thigh*, coefficient = −0.232, $p < 0.001$), the forearm flap's final anatomy is more responsive to the surgeon's corrective overbulking. We interpret this as a difference in "tunability": the compliant forearm flap is more malleable, allowing the added subcutaneous volume to be effectively shaped by the surrounding native tongue, leading to better anatomical conformity. This effect is reinforced by the *overbulking* × *stiffness_level* interaction (coefficient = 0.160, $p < 0.001$), which reveals that for any given donor site, the anatomical return on investment from overbulking is greatest when using the most pliable tissue available. This provides a direct, physics-based rationale for the clinical preference for the beavertail modification of the RFFF, which increases the subcutaneous bulk while maintaining the flap's overall pliability [10,55,56]. Furthermore, the significant *overbulking* × *Case* interactions highlight that the benefit of adding volume is not universal; it is dictated by the unique geometrical constraints of the resection site. Taken together, the model demonstrates that while overbulking is universally beneficial for restoring anatomy, its clinical effectiveness is highly variable and depends on the flap's inherent biomechanics and the specific demands of the defect.

Together, the surgical decisions of donor site selection and flap overbulking are not independent variables but a coupled system with compounding effects. The choice of donor site establishes a baseline biomechanical cost in the form of tissue strains and tensions. Moreover, it dictates the therapeutic effectiveness of overbulking; a pliable forearm flap imposes less initial strain and allows its final anatomy to be more effectively "tuned" with added volume. Conversely, a stiffer thigh flap introduces a higher baseline strain and shows diminished anatomical returns from the same corrective overbulking. Therefore, an optimal reconstructive strategy must consider these decisions in unison, recognizing that the inherent properties of the chosen flap define the potential and the limitations of subsequent efforts to restore anatomical form. This interdependence underscores the complexity of surgical planning and highlights the need for patient-specific models such as the one presented herein to navigate this complex decision-making process.

The presented framework introduces a novel approach for simulating the complex procedure of free flap reconstruction. This is achieved by modeling the flap as an independent, multi-component body that is virtually sutured to the resection site. This approach fundamentally advances two established paradigms in surgical simulation. First, it moves beyond the "property modification" approach, where the outcome of a reconstruction is simulated by altering the material properties of elements within a single, continuous mesh. Foundational work by Buchaillard et al. [23], later adopted by van Alphen et

al. [57] and more recently Al-Zanoon et al. [32], effectively used this simplification to explore the functional consequences of flap stiffness mismatch or fibrotic tissue. Our framework, in contrast, models the flap as a distinct biomechanical entity. This allows explicit capture of the physics of the flap-tongue interface, the induced deformations by suture tensioning, and the final post-operative equilibrium state. In essence, it simulates the surgical process, not just its endpoint material state. Second, our work extends the "primary closure" simulation paradigm, pioneered by Kappert et al. [37]. Their method represented a major advance by computationally replicating the surgical act of suturing, successfully modeling the deformation caused by pulling native tissue edges together to close a small defect. Our framework adapts this concept to the more complex and common scenario of microvascular free flap reconstruction.

Another key innovation of this pipeline is the simulation of post-operative tissue atrophy, which enables the model to evolve from its immediate "Day 0" state to a predicted one-year post-remodeling anatomy; the state most relevant to long-term patient function. By modeling this crucial remodeling phase, the framework provides a biomechanical basis to investigate the clinical practice of flap overbulking. The focus on long-term biological change complements prior simulation efforts that have also investigated post-treatment tissue alteration. Al-Zanoon et al. [32], for instance, modeled the development of radiation-induced fibrosis by increasing the stiffness of affected tissue elements. Our approach is complementary, modeling atrophy via a prescribed volumetric change, another critical remodeling phenomenon that dictates the final tongue morphology. Furthermore, while Wang et al. [58] specifically studied the functional consequences of base-of-tongue volume loss on swallowing, their work focused on the hydrodynamics of bolus flow driven by a prescribed boundary motion. Our framework, in contrast, simulates the mechanics of how the reconstructed neotongue itself deforms and settles into a new passive equilibrium as the flap tissue atrophies. This provides direct insight into the final resting anatomy and the resulting strains imposed on the native tongue, a different set of biomechanical outcomes.

The successful execution of a 120-simulation factorial study demonstrates the framework's core design principles: automation and robustness. This scalability marks a methodological shift. We present a high-throughput virtual testbed that contrasts with a significant body of research focused on building baseline models of the highest fidelity. Such landmark efforts include the comprehensive FRANK model, which integrates numerous head and neck components [25–27], and patient-specific models derived from DW-MRI that incorporate detailed muscle fiber architecture [31]. Our framework, however, makes a deliberate, pragmatic trade-off. Since our primary goal is to predict the final, passive anatomy at rest after biological remodeling, we do not model the active muscle architecture, which is critical for functional simulation but not for this specific question. Instead, the framework employs well-established isotropic hyperelastic models [28] rather than more complex anisotropic formulations, which have proven exceptionally difficult to parameterize from in vivo data [31]. This trade-off prioritizes simulation scalability over baseline complexity. The focus is therefore not on perfecting the pre-operative model but on robustly simulating the post-operative consequences that arise from interdependent surgical decisions. Ultimately, this capability distinguishes our approach from prior tongue surgery simulations, which have necessarily focused on single-case analyses or a small number of scenarios [23,37]. It is this scalability that opens the door to systematic, multi-parameter investigations of surgical variables.

The qualitative robustness of these findings extends to the specific choice of constitutive formulation. While this study utilized the Yeoh and Ogden models (selected for their efficacy in capturing the nonlinear stiffening of oral tissues [28]), alternative hyperelastic potentials would likely yield similar trends due to shared fundamental physics. Specifically, the reported trade-offs are mechanically driven by quasi-incompressibility and monotonic strain-stiffening; conservation of volume dictates that overbulking a confined space must invariably impose strain, while the universal convexity of valid soft tissue models ensures that the relative ranking of flap stiffness effects remains consistent regardless of the specific strain energy function employed.

This degree of automation and robustness is what transforms the framework from a research platform into a viable tool for patient-specific pre-operative planning. The laborious, manual, and expert-driven process of creating high-fidelity, patient-specific biomechanical models has traditionally been a major bottleneck, limiting their scalability for large-scale or

clinical applications [27,29–31]. In contrast, our framework is designed for clinical tractability. The pipeline requires only two triangulated meshes: the baseline tongue and the planned resection, which directly align with the modern VSP workflows where a surgeon delineates the resection on medical images. From this point, the entire workflow (from biomechanically optimized flap design to volumetric FEM generation and final post-atrophy analysis) is fully automated. The clinical necessity for such a tool is underscored by the results of the factorial study itself. The significant *overbulking* × *Case* interaction demonstrates that the optimal surgical strategy is dictated by the unique geometric constraints of the patient's specific defect. The streamlined and robust nature of the pipeline makes it computationally feasible to conduct a tailored series of simulations for an individual patient to optimize a critical variable, such as the overbulking percentage. This offers a direct, physics-based method to navigate the complex, interdependent surgical decisions, and move toward data-driven, individualized surgical planning.

The current framework, while robust, relies on a simplified representation of the tongue's complex biomechanics and post-operative healing. Future work should incorporate muscle fiber orientation data, potentially from DT-MRI, to capture the tongue's anisotropic properties. A key advancement will be the integration of active muscle contraction models, enabling the simulation of dynamic functions like speech and swallowing. Furthermore, the biological remodeling simulation can be enhanced by implementing patient-specific atrophy models that factor in variables such as radiation dose, flap composition, and individual factors such as age, sex, and BMI. Such studies could also benefit from modeling the formation of stiffer, contracted scar tissue at the incision interfaces. Finally, refining the constitutive parameters through in vivo techniques, such as MR elastography for the tongue [59] and pre-operative mechanical testing of donor skin [47], will increase the model's predictive accuracy.

To translate this framework into a clinically viable tool, several validation and generalization steps are necessary. A prospective clinical study is needed to validate the simulated anatomical outcomes against long-term post-operative imaging from a cohort of patients. The model's generalizability must also be established by applying it to a diverse range of patient anatomies and tumor-specific resection geometries. The robust pipeline is also well-suited for large-scale in silico studies to systematically investigate surgical variables, such as by comparing the biomechanically optimized flaps used herein against conventional flap designs sourced from surgeons. Ultimately, this framework has the potential to serve as a research platform for systematically investigating surgical variables and as a tool for conducting patient-specific pre-operative simulations to compare optimal reconstructive strategies.

## Conclusion

This study establishes a robust and automated simulation pipeline capable of modeling key aspects of tongue reconstruction surgery and its long-term anatomical remodeling. This framework provides a novel platform to systematically investigate interdependent surgical variables that are otherwise challenging to isolate clinically. Our findings offer a quantitative mechanism for the delicate balance surgeons must strike between restoring volume and preserving mobility. Specifically, the simulations confirm that increasing bulk invariably incurs a biomechanical cost (increased experienced strain) which can lead to the tethering of the mobile tongue. We demonstrate that donor site selection and the degree of overbulking function as a coupled system: selecting a stiffer donor site, such as the thigh, not only imposes higher baseline mechanical strain but also limits the anatomical effectiveness of subsequent overbulking. This reinforces the clinical rationale for pliable flaps, such as the beavertail-modified forearm flap, to achieve anatomical conformity with a lower biomechanical penalty. Crucially, significant interactions between surgical strategy and the patient's unique defect geometry confirm that a universal approach is suboptimal. These results highlight the framework's potential for patient-specific pre-operative planning, allowing surgeons to computationally explore scenarios tailored to the individual patient. While future refinements should incorporate anisotropic material properties and active muscle modeling, this work lays the foundation for a powerful research platform. Following clinical validation, it promises to serve as a decision-support tool to help surgeons optimize and individualize reconstructive strategies.

## Supporting information

**S1 Fig. Mesh convergence analysis—Maximum effective strain.** maximum effective strain of the neotongue at the final post-remodeling equilibrium versus total mesh nodes. Each curve is one of 12 representative parameter combinations and is labeled with a three-letter code in the form [Case][Donor/Stiffness][Overbulking], where Cases A–D are the four defect geometries; Donor/Stiffness A–C = forearm (soft→stiff), D–F = thigh (soft→stiff); Overbulking A–E = 0%, 22.2%, 44.4%, 66.7%, 88.9%.
(TIF)

**S2 Fig. Mesh convergence analysis—Maximum displacement.** maximum nodal displacement of the neotongue at the final post-remodeling equilibrium (mm) versus total mesh nodes for 12 representative parameter combinations (same [Case][Donor/Stiffness][Overbulking] codes as in S1 Fig). Two panels are shown for better readability; horizontal axis is shared.
(TIF)

## Author contributions

**Conceptualization:** Amir Reza Isazadeh, Lindsey Westover, Hadi Seikaly, Daniel Aalto.

**Data curation:** Amir Reza Isazadeh.

**Formal analysis:** Amir Reza Isazadeh, Daniel Aalto.

**Funding acquisition:** Daniel Aalto.

**Investigation:** Amir Reza Isazadeh, Lindsey Westover, Hadi Seikaly.

**Methodology:** Amir Reza Isazadeh, Lindsey Westover, Daniel Aalto.

**Project administration:** Amir Reza Isazadeh.

**Resources:** Daniel Aalto.

**Supervision:** Lindsey Westover, Hadi Seikaly, Daniel Aalto.

**Validation:** Amir Reza Isazadeh.

**Visualization:** Amir Reza Isazadeh.

**Writing – original draft:** Amir Reza Isazadeh.

**Writing – review & editing:** Amir Reza Isazadeh, Lindsey Westover, Hadi Seikaly, Daniel Aalto.

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
