## [Decision Letter · Decision Letter 0]

15 Jan 2026

Dear Dr. Isazadeh,

Thank you for submitting your manuscript to PLOS ONE. After careful consideration, we feel that it has merit but does not fully meet PLOS ONE’s publication criteria as it currently stands. Therefore, we invite you to submit a revised version of the manuscript that addresses the points raised during the review process.

Since one of the reviewers suggested the inclusion of additional bibliographic references, the authors are invited to assess the relevance of the suggested studies and to decide whether their inclusion is appropriate. The inclusion of the suggested citations remains at the authors’ discretion and is not mandatory if they are judged not to be pertinent to the manuscript.

We look forward to receiving your revised manuscript.

Kind regards,

Alice Berardo

Academic Editor

PLOS One

Journal Requirements:

Reviewers' comments:

Reviewer's Responses to Questions

**Comments to the Author**

1. Is the manuscript technically sound, and do the data support the conclusions?

Reviewer #1: Yes

Reviewer #2: Yes

2. Has the statistical analysis been performed appropriately and rigorously?

Reviewer #1: Yes

Reviewer #2: N/A

3. Have the authors made all data underlying the findings in their manuscript fully available?

Reviewer #1: Yes

Reviewer #2: Yes

4. Is the manuscript presented in an intelligible fashion and written in standard English?

Reviewer #1: Yes

Reviewer #2: Yes

Reviewer #1: The authors present a comprehensive work that provides a deep evaluation, through numerical simulations, of different scenarios and aspects related to tongue reconstruction surgery, with a particular focus on the donor site and flap volume. The topic is relevant, but clarifying certain methodological aspects and adding further discussion would enhance the manuscript before publication.

1 INTRODUCTION

- The authors state that tongue mobility after reconstruction is the most important aspect to preserve, and they describe and compare the characteristics of two donor sites: the Radial Forearm Free Flap and the Anterolateral Thigh. To provide better context for the study, it would be useful to provide a brief overview of the donor site choices commonly used in the literature, along with their pros and cons. This would highlight the advantages of the Radial Forearm and Anterolateral Thigh Free Flaps chosen for investigation and help readers understand their relevance to surgical outcomes.

2 METHOD

2.1 Patient-Specific Tongue Model

- The tongue model was derived from an MRI scan of a healthy male participant in his 40s. Including participant characteristics, such as height and/or BMI, would help better understand the model's tongue dimensions and those of the free flap under consideration.

- You state that: “To ensure consistency across representative cases, we adopted the same four clinically plausible tongue reconstruction scenarios and used ”resection models” identical to those in our prior work”. Please provide the reference for the work you are referring to and briefly describe the four scenarios for clarity.

- The patient-specific tongue model comprises the tongue, the resected tongue and the jaw. Please specify which software and procedures were used to extract the tongue model from MRI, obtain the resected tongue (through Boolean subtraction) and construct the rigid jaw models.

- Figure 1: It would improve the readability to indicate the letter of the four different panels (A-B) in the figure captions. In addition, please define the hiFEM acronym in the figure caption or within the text, explain why it is used, and describe how the planar flap design was obtained.

2.2 Biomechanically Optimized Flap Design

- Figure 2A is not cited in the main text; please include an explicit reference to it.

- Given the procedural workflow, in my opinion, it may improve the clarity to invert the order of panels B and C in Figure 2.

- Figure 2: Panels F and G represent the anterior and posterior views of the model; this should be specified in the figure caption to improve clarity.

2.3 Simulation of Tongue Reconstruction Surgery

- Please explicitly state in this section the software (FEBio) adopted to perform the computational simulation of the surgical procedure for tongue reconstruction.

- The author states that "Following the virtual suturing, a subsequent step allows the system to equilibrate for five minutes". Please clarify the rationale for selecting this time interval. Was the choice based on previous studies, experimental tests, or literature evidence?

2.6 Constitutive Models

- In the manuscript, it is reported that “All soft tissues (tongue, skin, and subcutaneous adipose tissue) were modelled as nearly incompressible, non-linear hyperelastic materials.” In my opinion, given the description of the tissue properties in the paragraph (the choice of Yeoh and Ogden models for the hyperelastic component and the introduction of viscous behaviour), it would be more accurate to state that “All soft tissues (…) were modelled as nearly incompressible, isotropic, non-linear hyperelastic and time-dependent materials”.

- Considering the parameters reported in Table 1, the constitutive models adopted to describe the behaviour of the tissues are a 2-term Yeoh model for the native tongue, a 2-term Ogden model for the skin, and a 1-term Ogden model for the adipose tissue. Since not all models include 2 terms, it may be clearer in equations (1) and (2) to report the summation over i from 1 to N and then specify the values of N for each tissue.

2.7 Soft Tissue Parametrization

- The authors report that, to account for the high experimental variability in tissue stiffness for both Forearm and Thigh skin, a boundary approach was adopted, resulting in three representative uniaxial stress-stretch curves for each donor site. However, the rationale for this approach and the criteria used to select the three reported curves are not clearly described. Please provide additional details to clarify this choice.

- Figure 4: The stiffness levels adopted for both the forearm and thigh skin are three; however, while all three curves are shown for the forearm, only two curves are reported for the thigh. Please also report the three representative uniaxial stress-stretch curves for the thigh site. In addition, the three stiffness-level curves for the forearm are difficult to distinguish; consider using, e.g., different colours or line styles. Finally, the x-axis is labelled as “Steretch”; please correct it to “stretch”.

- The tissue properties were incorporated into the simulations, adopting a 2-term Ogden model calibrated to the three different stiffness levels (both for the forearm and thigh). Please describe in more detail how this calibration was performed, specifying which method/optimisation procedure was adopted.

- Depending on the tissue, the number of viscous branches in the Prony series varies (4 for the native tongue and skin, 3 for the subcutaneous adipose tissue). Please specify the reason for this choice. Was it based on the literature or on the fitting/calibration procedure?

- Table 1: for the subcutaneous adipose tissue, the parameter m1 should be dimensionless. Please verify and correct it.

2.8 Simulation Stability and Robustness

- The results of the mesh convergence analysis are reported in the Supplementary Material. However, the number of nodes and elements used in the models (e.g., average values or a range) should also be reported in the section for completeness.

2.9 Outcomes Metrics

- The authors referred to panels v and vi of Figure 3. To improve clarity, it may be better to split this figure into two separate figures: Figure 3 with panels i-iv and Figure 5 with panels v and vi. In addition, it should be specified in the Figure, near each contour, which metric is under consideration: effective strain or Hausdorff distance.

3-4 RESULTS AND DISCUSSION

- Given that computational simulations are a crucial part of the study, it would be appropriate to include a discussion—and at least one representative figure—showing the main contours of effective strain and Hausdorff distance. While reporting all 120 simulations is not feasible, selected representative cases (e.g., specific reconstruction scenarios, tissue properties, and levels of overbulking) would be highly informative and enrich the manuscript.

- The study considers three stiffness levels at the two donor sites: the forearm and the thigh. Although stiffness is included as a factor in the regression analyses, the manuscript does not explicitly report how changes in stiffness alone affect the output metrics (effective strain and Hausdorff distance). Discussing the effects of stiffness on these metrics would strengthen the paper.

- Part of the study relies on a specific constitutive formulation; it would be useful for the Discussion to more explicitly comment on how the results might qualitatively change—or remain robust—if alternative constitutive models were adopted.

Reviewer #2: This study presents a robust, automated, and open-source computational testbed for evaluating interdependent surgical variables in free flap tongue reconstruction. By quantifying anatomy–biomechanics trade-offs through large-scale simulation, the framework provides physics-based insights into the consequences of donor site selection, flap stiffness, and overbulking. Its scalability and automation offer a pathway toward personalized, simulation-informed surgical planning, with the potential to improve functional outcomes in tongue reconstruction.

Before the Editor makes a decision, I suggest that the authors must take into account the following corrections:

1. I think the title needs to be reformulated to become more "friendly".

2. The present form of the abstract is a bit week not much clear. Hence, I recommend to re-write it with 3/4 stronger sentences about your objectives/ findings that will give a better understanding for the readers

3. The "Introduction" section should be more concise and some sentences should be rewritten.

4. The manuscript is well written, but the Introduction is very concise, and more articles and works need to be discussed and explained. Some other related and interesting topics need to be cited as well, as below:

- "Effect of thermal dispersion on free convection in a fluid saturated porous medium," International Journal of Heat and Fluid Flow, 30, 229-236 (2009). http://dx.doi.org/10.1016/j.ijheatfluidflow.2009.01.004

-Analytical estimation of temperature in living tissues using the TPL bioheat model with experimental verification. Mathematics, 8(7) (2020) http://dx.doi.org/10.3390/math8071188.

-Thermal response of cylindrical tissue induced by laser irradiation with experimental study. International Journal of Numerical Methods for Heat and Fluid Flow, 22(7), 4013-4023 (2020) http://dx.doi.org/10.1108/HFF-10-2019-0777.

-Analytical solutions of thermal damage in living tissues due to laser irradiation. Waves in Random and Complex Media, 31(6), 1443-1456 (2021) http://dx.doi.org/10.1080/17455030.2019.1676934.

-An analytical solution of the bioheat model in a spherical tissue due to laser irradiation. Indian Journal of Physics, 94(9), 1329-1334 (2020) http://dx.doi.org/10.1007/s12648-019-01581-w.

- Effects of magnetohydrodynamic flow past a vertical plate with variable surface temperature. Applied Mathematics and Mechanics (English Edition), 31(3), 329-338 (2010) http://dx.doi.org/10.1007/s10483-010-0306-9

5. The paper is well written and serves the purpose. Except some minor typographical errors there are no major issues.

6. The conclusion must be improve and some sentences should be rewritten.

what does this mean?). If published, this will include your full peer review and any attached files.

**Do you want your identity to be public for this peer review?** For information about this choice, including consent withdrawal, please see our Privacy Policy

Reviewer #1: No

Reviewer #2: No

---

## [Author Response · Author response to Decision Letter 1]

20 Feb 2026

Detailed Response: "Response-Letter-PONE-D-25-59556.pdf"

---

## [Decision Letter · Decision Letter 1]

10 Mar 2026

Computational analysis of tongue reconstruction surgery: The impact of donor site and flap volume on post-operative anatomy and biomechanics

PONE-D-25-59556R1

Dear Dr. Isazadeh,

We’re pleased to inform you that your manuscript has been judged scientifically suitable for publication and will be formally accepted for publication once it meets all outstanding technical requirements.

Kind regards,

Alice Berardo

Academic Editor

PLOS One

Additional Editor Comments (optional):

As one of the two reviewers was no longer available to evaluate the revised manuscript, I reviewed the revised version myself following the first round of peer review.

Reviewers' comments:

Reviewer's Responses to Questions

**Comments to the Author**

Reviewer #1: All comments have been addressed

2. Is the manuscript technically sound, and do the data support the conclusions?

Reviewer #1: Yes

3. Has the statistical analysis been performed appropriately and rigorously?

Reviewer #1: Yes

4. Have the authors made all data underlying the findings in their manuscript fully available?

Reviewer #1: Yes

5. Is the manuscript presented in an intelligible fashion and written in standard English?

Reviewer #1: Yes

Reviewer #1: The authors have adequately addressed all the concerns raised in the first round of revision. The manuscript has significantly improved and is now stronger overall.

I believe that providing some additional details regarding the model calibration, particularly the method and optimisation procedure adopted, would have been beneficial; however, this remains the only minor point.

what does this mean?). If published, this will include your full peer review and any attached files.

**Do you want your identity to be public for this peer review?** For information about this choice, including consent withdrawal, please see our Privacy Policy

Reviewer #1: No

---

## [Editor Report · Acceptance letter]

PONE-D-25-59556R1

PLOS One

Dear Dr. Isazadeh,

I'm pleased to inform you that your manuscript has been deemed suitable for publication in PLOS One. Congratulations! Your manuscript is now being handed over to our production team.

Kind regards,

on behalf of

Dr. Alice Berardo

Academic Editor

PLOS One